# Drones for Conservation in Protected Areas: Present and Future

**Jesús Jiménez López [1,\*] and Margarita Mulero-Pázmány [2,\*]**

[1]  MARE—Marine and Environmental Sciences Centre, Quinta do Lorde Marina, Sítio da Piedade, 9200-044 Caniçal, Madeira Island, Portugal

[2]  School of Natural Sciences and Psychology, Liverpool John Moores University, Liverpool L3 3AF, UK

\*  Correspondence: lopezjimenezjesus@mare-centre.pt (J.J.L.); M.C.MuleroPazmany@ljmu.ac.uk (M.M.-P.)

**Abstract:** Park managers call for cost-effective and innovative solutions to handle a wide variety of environmental problems that threaten biodiversity in protected areas. Recently, drones have been called upon to revolutionize conservation and hold great potential to evolve and raise better-informed decisions to assist management. Despite great expectations, the benefits that drones could bring to foster effectiveness remain fundamentally unexplored. To address this gap, we performed a literature review about the use of drones in conservation. We selected a total of 256 studies, of which 99 were carried out in protected areas. We classified the studies in five distinct areas of applications: "wildlife monitoring and management"; "ecosystem monitoring"; "law enforcement"; "ecotourism"; and "environmental management and disaster response". We also identified specific gaps and challenges that would allow for the expansion of critical research or monitoring. Our results support the evidence that drones hold merits to serve conservation actions and reinforce effective management, but multidisciplinary research must resolve the operational and analytical shortcomings that undermine the prospects for drones integration in protected areas.

**Keywords:** protected areas; drones; RPAS; conservation; effective management; biodiversity threats

## 1. Introduction

Protected areas aim to safeguard biodiversity, preserve ecosystem services and ensure the persistence of natural heritage [1]. Despite their essential role in conservation, the allocation of resources to cope with an increasing variety of regular activities and unforeseen circumstances remains generally insufficient [2], severely affecting overall effectiveness [3]. Besides, protected areas subjected to international and national agreements must resolve their acquired responsibilities to maintain their legal status [4]. Hence, there is a demand for cost-effective, versatile and practical initiatives to attend a disparity of requirements to guarantee conservation, including a wide range of natural solutions [5], technological advances, and methods or innovative application of existing technologies [6].

In the last decade, drones (also known as unmanned aerial systems, remotely piloted aircraft systems, RPAS, UAS, UAV) have been the subject of a growing interest in both the civilian and scientific sphere, and indeed avowed as a new distinct era of remote sensing [7] for the study of the environment [8]. Drones offer a relatively risk-free and low-cost manner to rapidly and systematically observe natural phenomena at high spatio-temporal resolution [9]. For these reasons, drones have recently become a major trend in wildlife research [10,11] and management [12–14].

The success of drones can be partially explained by their great flexibility to carry different sensors and devices. The scope of application determines the best combination of aerial platform and payload. Although drones come in many different shapes and sizes, widespread small fixed-wing and rotary-wing aircrafts are frequently used for video and still photography. These consumer grade

drones coupled with lightweight cameras and multispectral sensors can deliver professional mapping solutions at a fraction of a cost than previous photogrammetric techniques. Medium size drones can be equipped with compact thermal vision cameras, hyperspectral sensors and laser scanning such as LiDAR, with great prospects for wildlife ecology, vegetation studies and forestry applications respectively [15–17]. Even though visible and multispectral band cameras encompass the most obvious sensing devices, drones can indeed incorporate a diversity of instruments to measure many distinct physical quantities such as temperature, humidity or air pollution [18]. Additionally, large aerial platforms can lift heavier payloads and represent an appropriate solution for integrating complex systems with the capacity to remotely assist sampling, hold cargo or deliver assistance. A brief summary of platforms and sensors is given in Tables 1 and 2 (but see [19–22] for an in-depth revision).

**Table 1.** Classification of drones according to characteristics and applications.

| SIZE | | | | | |
|---|---|---|---|---|---|
| Nano <br> <30 mm | Micro <br> 30–100 mm | Mini <br> 100–300 mm | Small <br> 300–500 mm | Medium <br> 500 mm–2 m | Large <br> >2 m |
| **Maximum Take-Off Weight (MTOW)** | | | | | |
| <0.5 Kg | 0.5–5 Kg | | 5–25 Kg | | >25 Kg |
| **RANGE (Distance/Type of Operation)** | | | | | |
| Close-range <0.5 miles | | Mid-range 0.5–5 miles | | Long-range 5 > miles | |
| Visual Line Of Sight (VLOS) | | Extended Visual Line Of Sight (EVLOS) | | Beyond Visual Line Of Sight (BVLOS) | |
| **WING** | | | | | |
| Rotary wing | | | | Fixed wing | |
| Single Dual rotors | Multi-Rotor | | | Low Wing / Mid Wing / High Wing / Delta Wing | Hybrid (VTOL) |
| | Tricopter | Quadcopter | Hexacopter / Octocopter | | |
| **POWER** | | | | | |
| Electric | | Gas | | Nitro | Solar |
| **ASSEMBLING** | | | | | |
| Ready-To-Fly (RTF) | | Bind-N-Fly (BNF) | | Almost-Ready-to-Fly (ARF) | |
| **APPLICATIONS** | | | | | |
| Logistics | Civil Engineering | Disaster Relief | Heritage | Search and Rescue / Precision Agriculture | Natural Resources | Law Enforcement |
| Wildlife Management | Weather Forecasting | Industrial Inspection | Leisure | Military / Disaster Relief | Aerial Photography and Film | Archeology |

Note: SIZE, MTOW and RANGE: based on average values (no specific standard/regulation). ASSEMBLING: level of work required to use the drone since acquisition.

Considering the ample range of possibilities, it is not surprising that some protected areas are adopting drones for various applications. For example, to assist search and rescue [23]; protect endangered turtles from feral species [24]; monitoring invasive plant species [25]; document illegal logging and mining [26]; wetland management [27]; anti-poaching [28]; and marine litter detection [29]. Recently, a team of scientists discovered a biodiversity hotspot using drones [30], which could be argued as a convenient procedure to adequately expand protected areas as established by the Aichi Target 11 [3]. In addition, we are witnessing a continuous development of sophisticated drones and ingenious methods that target particular conservation actions, such as wildfires firefighting [31]; whale health monitoring [32]; disease vectors control [33]; or seed planting for habitat restoration [34]. The fast pace of technological advances and novel applications probably exceeded previous expectations, but also gives rise to singular circumstances that must be placed in the context of management.

**Table 2.** Summary classification of sensors and devices that can be coupled to drones.

| Instrument. | | Type of Sensor | Spatial Resolution | Spectral Resolution | Weight | Costs |
|---|---|---|---|---|---|---|
| Imaging sensors | Visible RGB | Passive | Very high 1–5 cm/pixel | Low (3 bands) | Low <0.5 kg | Low $100–1000 |
| | Near Infrared (NIR) | Passive | Very high 1–5 cm/pixel | Low (3 bands) | Low <0.5 kg | Low $100–1000 |
| | Multispectral | Passive | High 5–10 cm/pixel | Medium (5–12 bands) | Medium 0.5–1 kg | Medium $1000–10,000 |
| | Hyperspectral | Passive | High 5–10 cm | High (> 50–100 bands) | Medium 0.5–1 kg | High $10,000–50,000 |
| | Thermal | Passive | Medium 10–50 cm/pixel | Low 1 band | Medium 0.5–1 kg | Medium $1000–10,000 |
| Ranging sensors | Laser scanners (LiDAR) | Active | Very high 1–5 cm/pixel | Low 1–2 bands | High 0.5–5 kg | High $10,000–50,000 |
| | Synthetic Aperture Radars (SAR) | Active | Medium 10–50 cm/pixel | Low 1 band | High >5 kg | Very high >$50,000 |
| **Other sensors and devices** | | | | | | |
| Atmospheric sensors | Temperature, Pressure, Wind, Humidity | | | | | |
| Chemical Sensors | Gas, Geochemical | | | | | |
| Position systems | Ultrasound, Infrared, Radio Frequency, GPS | | | | | |
| Other devices | Recorder device/microphones | | | | | |
| Sampling Devices | Water, Aerobiological, Microbiological Sampling | | | | | |
| Other devices | Cargo, Spraying, Seed spreader | | | | | |

Some authors have identified negative aspects of drones use in conservation. Potential wildlife disturbance effects [35] need to be further investigated. The use of drones as tools of coercion could weaken the environmental commitment of communities in protected areas [36], and therefore may prove counterproductive for conservation. On the other hand, the massive amount of data acquired with drones require modern, robust and computationally intensive methods to derive accurate and meaningful information [37], which may represent a technological barrier to the effective use of this technology in protected areas.

Likewise, the connection of drone advances with the most important features guiding effective management has not yet been specifically weighted and would be necessary to better align research efforts to conservation priorities. In addition, whether decision makers can take practical advantage of present and oncoming advances in the discipline remains questionable for several reasons. To find early answers to these remarks, we conducted an extensive literature review of drone applications with potential to enhance the effective management of protected areas. This perspective may help identify plausible scenarios where drones can be used in a rational and efficient manner.

## 2. Methods

We conducted a comprehensive literature search on drones in conservation up to October 2nd 2018, in line with related studies [10,11,35]. All searches were done by the same person in English, mainly using Google Scholar. This was further complemented through reference harvesting, citation tracking, abstracts in conference programs, and author search, using Research Gate and Mendeley (see PRISMA Flowchart in Supplementary Figure S1 Checklist and list of studies reviewed in Table S1). We then removed duplicate and unrelated results. Finally, peer-reviewed publications were collated and revised.

Keywords on the search included drones in their various meanings and acronyms: "unmanned aircraft systems", "UAS", "remotely piloted aerial system", "RPAS", "drone", "model aircraft",

"unmanned aerial vehicle", "UAV", "unmanned aircraft system". These were combined with terms referring to threats and common conservation measurements in protected areas: "protected area", "conservation", "ecology", "ecosystem", "habitat", "vegetation", "forest", "wetland", "reforestation", "monitoring", "survey", "sampling", "inventory", "wildlife", "fauna", "bird", "mammal", "fish", "amphibian", "reptile", "wildfire", "landslide", "remote sensing", "tourism", "ecotourism", "law enforcement", "poaching", "anti-poaching", "logging", "risk management", "pollution", and "search and rescue". In total, we applied 47 search terms and combinations using logical disjunctions.

We classified the studies into categories that represent the common threats and essential management measures in protected areas [5,38–40]. The categories are: "wildlife research and management" for those projects aimed at observing wildlife, estimating population parameters such as abundance and distribution, and establishing management measures to mitigate human-wildlife conflicts (n = 96); "ecosystem monitoring" for applications related with the study and mapping of natural habitats (n = 106); "Law enforcement" encompassing poaching and other illicit activities (n = 6); "Ecotourism" referring to recreational activities and visitors management (n = 3); "Environmental management and emergency response" spanning environmental monitoring and protection, natural hazards, search and rescue operations and similar cases (n = 45). We briefly tackled legal and ethical issues, including potential impact on wildlife and habitats, but also economic and technological factors, since all shape the feasibility of drones to approach conservation and environmental issues.

## 3. Results and Discussion

The literature search on drones in conservation provided a total of 256 studies. Of these, 99 describe applications that were accomplished in terrestrial and marine protected areas, according to the Protected Planet database [41]. The typology of protected areas includes national, international designations and registered private initiatives, with all UICN management categories (Ia, Ib, II, III, IV, V, VI) represented [1]. We found examples on all continents and in most ecosystems. The United States of America lead the ranking of countries where more drone studies have taken place (45), followed by Canada (26), Australia (17), China (11), Germany (11) and Spain (9). Figure 1 summarizes the selected research.

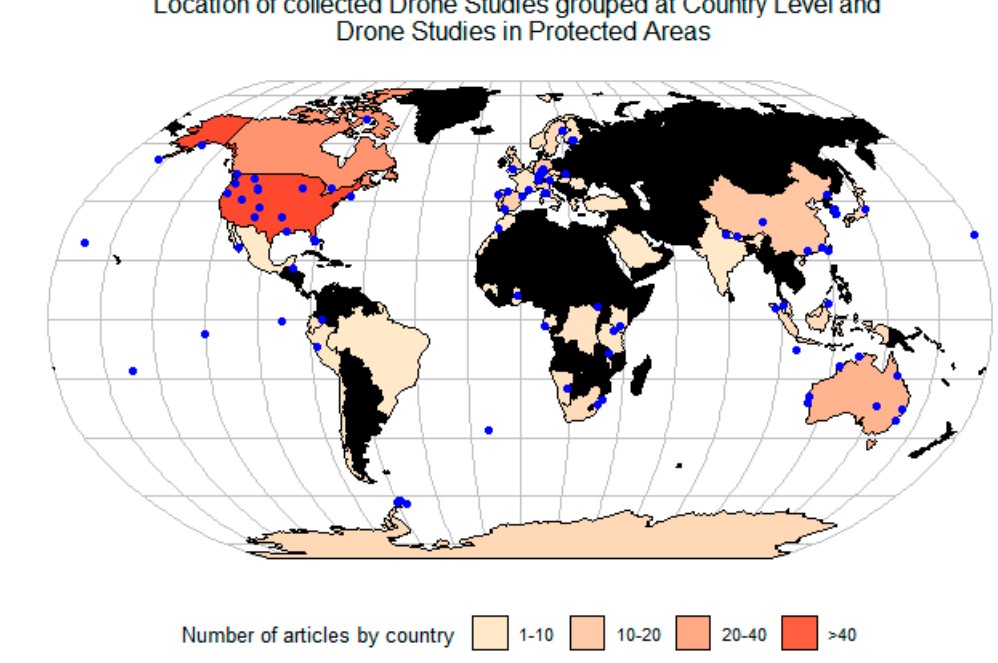

**Figure 1.** *Cont.*

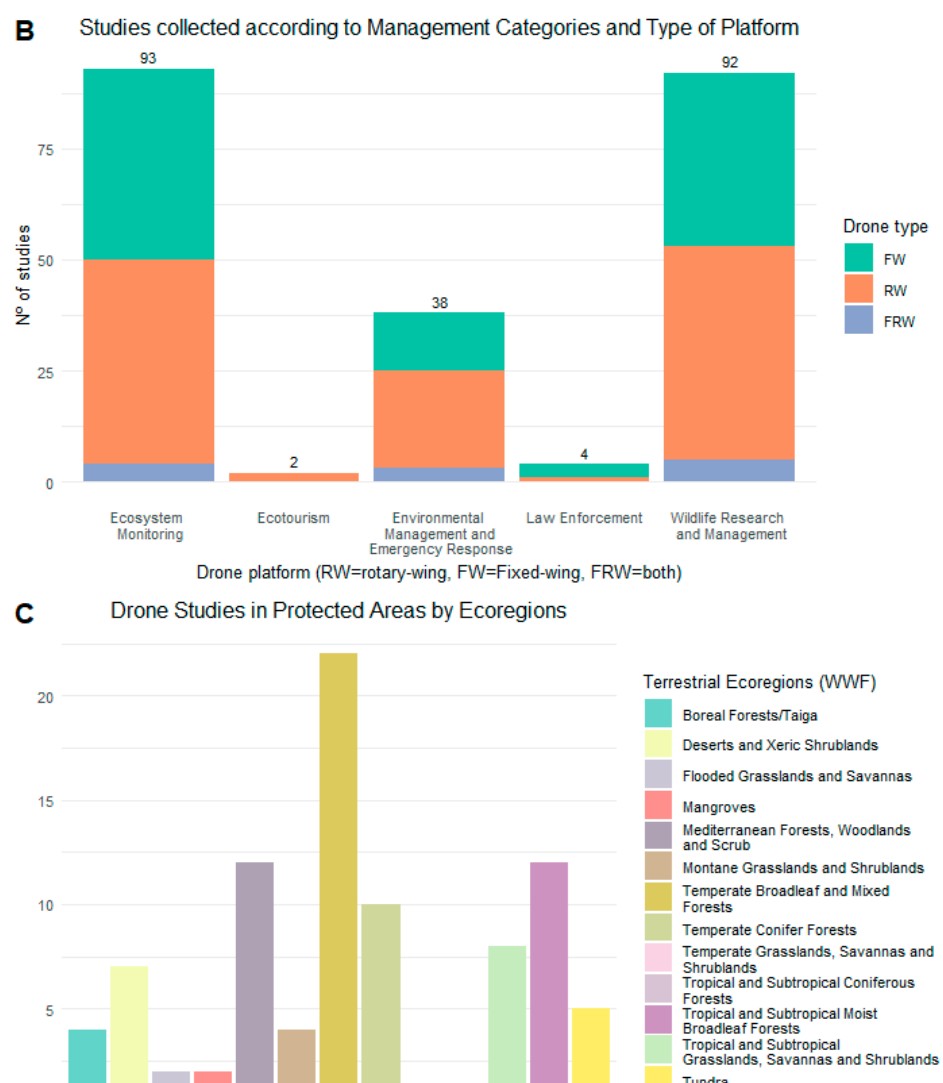

**Figure 1.** (**A**) Blue points represent studies in protected areas. Choropleth map shows location of studies by country. No studies were collected in countries colored black. (**B**) Only studies where type of platform was identified are shown. (**C**) Information extracted from WWF Terrestrial Ecoregions Map [42]. No drone studies were found in Protected Areas with Tropical and subtropical dry broadleaf forests.

The classification of the studies in categories that align with recurring aspects of conservation and management in protected areas [43] provides a framework that may help park-managers to identify feasible drone scenarios. The factors influencing effectiveness can be conveniently ascribed to the proposed categories and associated with consensual conservation actions [44]. In the next sections, we discuss the current state of the art and the challenges for the future integration of drones in protected areas.

*3.1. State of the Art: Drones in Protected Areas*

3.1.1. Wildlife Research and Management

Manned aircrafts have been traditionally used to complement ground-based wildlife surveys, but under-resourcing of many protected areas prevent their more widespread use. Besides, a significant number of aerial accidents with fatalities have been historically reported [45]. Moreover,

aerial incursions are subject to visibility bias since a greater number of observers is required to guarantee an exhaustive count of populations and minimize errors [46]. Drones have emerged as a feasible alternative to surpass such inconveniences at small scales and complement modern wildlife conservation. Remotely sensed capabilities of drones offer a less invasive, non-hazardous, repetitive and reliable monitoring technique [47] to collect species abundance and distribution, document wildlife behavior, life-history and health status. Recent examples target terrestrial mammals [48–50]; marine mammals [51–55]; birds [11,56–60]; reptiles [15,61–64]; and fish [65,66]. Most surveys opted for both optical and thermal cameras, the latter especially appropriate to sense elusive species overnight, when the temperature differences between the animal body and the environment are greater [67]. Other studies implemented acoustic sensors to record songbirds [68] or combine drones with tracking systems aboard [9,69,70] to collect wildlife movement and environmental data. Researchers have also devised ways to use drones for insect monitoring [71], habitat modeling [72] and sampling [73].

Protected areas often face human-wildlife conflicts in populated areas bordering their limits [74]. Some studies described the use of drones in various management tasks, such as moving elephants out of human settlements [75], mapping wildlife damage on crops to calculate compensation costs [76] or dropping fake baits targeting feral species [77]. Drones constitute an attainable low-cost alternative to assess and reduce the risk that hazardous infrastructures [78,79] or mechanical harvesting [80,81] pose to wildlife. Lastly, fine-scale mapping of species distribution, land-use changes and water bodies using high resolution aerial imagery hold potential to complement epidemiological and zoonotic studies [82–85], and may serve as a rapid mechanism to inform prevention and reinforce biosecurity programs.

3.1.2. Ecosystem Monitoring

Protected areas are reference sites for ecological monitoring. These activities provide essential information to track ecosystem changes as a result of management and environmental factors [86]. Established methods for habitat monitoring range from in situ and airborne observations to satellite-based remote sensing. The latest generation of commercial satellite sensors [87] collect images at sub-meter resolution and entail remarkable technological advances to Earth observation, but the geographical availability of products is limited and not always rapidly available. Drones are particularly appropriated to timely survey small areas at unprecedented detail [88], could be adapted to carry sampling devices and take in-situ measurements [89], and may prove advantageous to monitor Essential Biodiversity Variables (EBVs) [90]. Similarly, mapping and quantifying ecosystem services with drones constitute an efficient means to inform site design and zoning, especially when the information available is scarce, outdated and based on coarse-resolution remote sensing images. Also, monitoring habitat degradation with drones in protected areas and borderlands [91,92] represents a novel method to assess the performance of conservation actions. Finally, fine-scale habitat assessment using high resolution maps could assist, selecting suitable reintroduction sites for endangered or locally extinct species [93].

Experimental drone monitoring projects have increased noticeably, both by governmental institutions [94] and research groups, for informing on the distribution [95], health [16,96], productivity [97], composition [98], structure [99,100] and biomass [101–104] of forests using both passive and active sensors [105]. As a consequence, drone applications for inventory, characterization and habitat restoration are maturing fast, but scaling-up and linking the collected information with that coming from satellite remote sensing remains a knowledge gap [106]. However, some studies represent a step in this direction, including the following: derive and enhance ground-based forest metrics to assist modeling of ecological process at regional scale [107], validate vegetation maps from drone image interpretation [108,109] or address the radiometric calibration of small multispectral cameras to allow comparisons with satellite data [110,111]. Drones have been used for community-based forest monitoring [112], and therefore suggested as an important asset to impulse the participation of developing countries in the carbon market (Reducing Emissions from

Deforestation and Forest Degradation, REDD) [113]. In addition, drones have operated successfully in different ecosystems to measure the spread of invasive species [114–118]; map coastal and marine habitats [119–125]; wetlands [126–130]; grasslands [17,131,132]; savannas [133,134]; glaciers [135–137]; polar areas [138,139]; and riparian ecosystems [140–143].

### 3.1.3. Law Enforcement

Efficient control and surveillance of illegal activities lead the ranking of measures for effective management of terrestrial [144] and marine [145] protected areas. These conservation actions aim to maintain the integrity of threatened species and ecosystems in the face of human pressures, but in practice suffer from serious deficiencies [146]. Enforcement is especially challenging in large protected areas where iconic species are on the verge of extinction due to illegal hunting, fishing, encroachment or habitat loss. Drones constitute a technological advance to complement insufficient staff and resourcing in anti-poaching [12,147–149] and other less contentious acts such as vandalism or bonfires in unauthorized areas [150,151]. Drone surveillance aim to autonomously detect and track subjects integrating live streaming visible and thermal camera systems with real time vision processing techniques. However, these applications are subjected to technological and legal constraints. Real-time recognition of suspicious activity or flying in adverse weather conditions remain a work in progress [152]. The relatively low maximum flight time of modest drones is a major obstacle to cover large areas [12], but progress is noticeable. Although the last generation of long-endurance fixed-wing and hybrid aerial platforms have higher autonomy, meeting the optimal specifications requires a considerable investment [153] with uncertain benefits, especially in developing countries [154]. Besides, the main barriers to protected areas surveillance using drones take place in the legislative and socio-political sphere. The flight rules often limit flying drones beyond the visual line of sight (BVLOS), above a certain altitude or at night, precluding the surveillance in periods of increased illegal activity. On the other hand, there are concerns about the alleged social and ethical implications of using drones with coercive purposes [155]. Duffy debated the advent of militarized conservation and stated that drones and similar technologies could contribute to human rights breaching [156], which may lessen the commitment of native communities [36,157] to protect their natural resources. Under these considerations, more research is needed to identify those technological advances and best practices that do not pose or minimize the risk to the privacy and welfare of people but serve for the purpose of surveillance. In this sense, thermal images reveal the temperature profile of the target, but lack the ability to collect sensitive personal information. Other measures can be taken to restrict the surveillance to previously defined zones and according to poaching threat maps [158] representing those areas with greatest pressures. In addition, some studies have remarked that the effectiveness of antipoaching depends on a greater allocation of resources [144]. For example, to improve the effectiveness of offshore guarding activities [159], patrol vessel could acquire waterproof rotary-wing or fixed-wing drones with float planes to persuade and record illegal fishing within the boundaries of marine protected areas. These evidences could be considered a reliable proof in court, even when offenders are seized outside the no-take zones [160]. Alternatively, there are some reported experiences where drones assisted counter-mapping with reasonable success [161,162]. With all due caution, these are some compelling reasons to encourage the development and implementation of drones to fight poaching. Nevertheless, the success of such initiatives might require a greater consensus among the parties involved and the development of multidisciplinary strategies that seek to solve these recurrent threats to biodiversity.

### 3.1.4. Ecotourism

Well-managed ecotourism promotes conservation and provides socioeconomic benefits to local communities. Otherwise, it may adversely affect the welfare of the animals and disrupt their habitats [163]. In the midst of the dilemma, drones have been proposed for recreational and educational purposes [164,165], document natural monuments and cultural sites [166]; and social

research and visitor surveillance [167,168]. However, drone operations are susceptible to endanger wildlife [35], compromise tourist experience [153] or in case of accidents, lead to pollution or wildfires in sensitive areas due to the presence of toxic and flammable components. Subsequently, to restrain the uncontrolled presence of drones in protected areas, stakeholders agreed on a set of policies to establish permitted activities in Antarctica [169], opted for simpler rules and recommendations [170] or completely banned drones arguing safety reasons and wildlife impact [171]. Even when the economic benefits and leisure possibilities are promising, undesirable events and a lack of ethical practices could emphasize the negative connotations of drones to the detriment of their advantages. Thus, it would be advisable to be cautious in the face of a growing demand to incorporate drones into ecotourism services and continue working on a set of consensual measures to minimize the potential drawbacks drones may bring to protected areas.

### 3.1.5. Environmental Management and Disaster Response

Effectively managing protected areas requires continuous monitoring of environmental biophysical indicators to ensure that potential sources of contamination are controlled or below a safety threshold and, if necessary, take appropriate restoration measures. In many cases, a rapid response is crucial to diminish the effects that natural and man-made disasters pose to natural resources and human beings. Usually, these conservation actions combine fieldwork, airborne and satellite remote sensing. Drone capabilities provide a fine-scale alternative to remotely assist water, soil and air quality sampling [172–176], and enable rapid image acquisition to monitor erosion [177]; sediments dynamics [178,179]; forest windthrow [180]; habitat degradation [125]; landslides [181–183]; flood [184]; volcanic events [183,185,186]; oil spills [187]; and wildfires [188–190] at different stages. Drones may also serve as valuable tools for rangers in search and rescue missions in marine and remote mountainous regions [191,192]. Besides, there are a variety of plausible scenarios where drones can prove to be useful, such as detecting marine litter [193–195], inspect facilities [196]; collect information gathered from environmental sensor networks [197]; or support plant invasion monitoring [198] and control by means of aerially deployed herbicide on targeted species [199].

### *3.2. Current Challenges on the Integration of Drones in Protected Areas*

### 3.2.1. Legal Barriers and Ethical Constraints

Drone operations face important social and legal barriers that undermine their potential in the civilian sphere [36,200,201]. Not without founded reasons, an overly restrictive and indiscriminate regulatory framework arguing privacy and safety issues is currently limiting the applications of drones in the field of conservation. This highlights the urgent need to seek consensus among countries and adapt legislation to distinguish between the purpose of leisure, research and management [202].

### 3.2.2. Impact of Drones on Wildlife and Ecosystems

Animal welfare and alteration of sensitive habitat in wildlife management and ecological research is a source of strong debate [203,204]. Some authors have reported disturbance effects of drones on birds [57,205–209], reptiles [210] and mammals [211–213]. Despite a greater degree of awareness reflected in a emergent set of guidelines to minimize the impacts on wildlife [35,56,214,215], most studies marginally inform reactions and further trials aimed at quantifying changes in behavioral patterns and physiological effects targeting a broader group of species is recommended. An optimal trade-off between benefits and environmental costs should be weighed [216,217]. By designing quieter, non-polluting and safer components, along with following up the suggested flight patterns, the impact on wildlife and ecosystems could be reduced and its objective and unbiased observation facilitated [47,204]. Therefore, drones have great potential to evolve, replacing more invasive monitoring techniques. This should be consciously considered by those reluctant to integrate drones in research and conservation activities. Step by step, a code of best practice and recommendations

could be continuously updated based on lessons learned [206], forming the basis for wildlife certified drone operators [35].

### 3.2.3. Costs of Drone Operation

Expenses derived from using drones in the long term are difficult to quantify [218] and depend on a confluence of factors. Some of the applications described above rely on the acquisition of sophisticated on-board instruments, devices and sensors, advanced communications system or gas-powered engines for longer endurance and heavier payloads. The large volume of data collected must be conveniently stored and processed, which often require qualified staff and adequate IT (Information Technology) infrastructures. In addition, operations with drones are not exempt from accidents, which may compromise the viability of some projects. The payload is usually the most expensive part of the platform, and this often breaks down. Park managers should be aware that there is not a single solution covering all the conservation purposes [219] and a trade-off analysis among available platforms and sensors should be pondered. In this regard, do-it-yourself (DIY) drones can be equipped with a flexible array of sensors and according to very specific requirements, but extra time and experience is required for the correct assemblage and configuration of parts. Since ready to fly commercial platforms are tested and proven systems, it could be argued that they present more reliable capabilities than custom-built drones. Moreover, the consumer market shows a gradual drop on prices in higher performance platforms [220]. Suppliers often provide support, training and companion software, albeit services could be occasionally charged. Nonetheless, there is general agreement that costs associated with drones are lower compared to established methods (Table 3), such as manned aircraft and ground incursions [13,178,211], at least for mapping small and medium scale areas. Although the benefits of monitoring greater extensions with drones remain challenging according to the state of the art, the situation is likely to be more favorable with the advent of more efficient aerial platforms.

**Table 3.** Examples of studies reporting favorable use of drones compared with established methods.

| Study | Aim | Established Methods | Using Drones |
|---|---|---|---|
| [173] | Water Sampling | **Boat sampling**<br>• 3 scientists, 1 boat, 1 truck, 1 trailer.<br>• Slow, spatially restricted.<br>• Expensive and laborious deployment<br>• lake sampled/10–15-h day. | • 2 h, 1 scientist, 1 drone.<br>• Sample all lakes at very high spatio-temporal resolution. |
| [57] | Nesting status of birds | **Climbing trees**<br>• 2500$<br>• 2 people and climbing gear.<br>• 33 min/inspection | • 1000$<br>• 1 person and drone<br>• 4:30 min/inspection |
| [57] | Elasmobranchs densities | **Fishing methods, diver surveys, video cameras, aerial surveys**<br>• Potential invasive methods<br>• Prohibitive cost.<br>• Risk for observers and observer bias. | • <2500 $<br>• Short period of time.<br>• True densities |
| [61] | Crocodile nesting behavior | **Helicopter, airboat, ground surveys**<br>• Prohibitive cost.<br>• Dangerous incursions. | • Low cost, repeatability, and flexibility |
| [221] | Mangrove forest inventory | **Fieldwork**<br>• Laborious and costly<br>• Trade-off sample size and frequency<br>• Located in remote areas.<br>• Disturbance of fauna and flora | • Consumer-grade drone 1200 $<br>• Above ground biomass estimation.<br>• Increase sampling frequency<br>• Less invasive |

### 3.2.4. Technological Challenges

As previously noted, the massive volume of data that sensors collect in the course of the surveys need to be stored, processed and analyzed, causing severe procedural bottlenecks [6] that need to be solved. When using aerial images for wildlife census, the manual counting and identification of individuals represent a considerable investment in time and costs. Progress in computer vision and machine learning are intended to automate such routine tasks [52,81,222–228]. Despite encouraging results [229], these methods are only available for relatively easy to spot species in open natural environments and require highly qualified personnel to offer reliable results. In addition, further research is required to assess the overall performance of drone data collection techniques to address the analysis and modelling of species distribution, especially in comparison with more mature statistical and sampling methods [58]. On the other hand, traditional pixel-based algorithms are rather inefficient when processing very high resolution images [128]. Therefore, object-based image analysis (OBIA) and deep learning techniques [230] will likely prevail during the next generation of land-cover, habitat and vegetation classification methods [8]. The arrival of affordable hyperspectral miniaturized sensors [124,128,231,232], will bring more complexity to the matter, requiring novel analytical approaches not currently implemented. Conversely, the entire photogrammetric process is well documented [233] and supported by commercial desktop and mobile applications, but also open source solutions [21], probably at expense of a major level of expertise [234,235]. Drones using Real Time Kinematic (RTK) and Post Processing Kinematic (PPK) techniques can produce survey-grade maps without requiring labor intensive ground control points (GCPs). Yet, the radiometric calibration of aerial images requires additional improvements [37] since it is considered a crucial step to carry out multi-temporal studies [236]. The confluence of big data [237], networked drones [238,239], artificial intelligence and sensors will bring new unforeseen perspectives to conservation, but integration of products and services to deliver off-the-shelf management solutions are still in their infancy.

### 3.3. Linking Drone Platforms and Sensors with Conservation

Park managers considering the acquisition of drones may need expert guidance to select the most suitable platform and sensor for each purpose. Here we provide a brief summary of most common imaging and ranging sensors (Table 4). Consumer grade cameras are adequate for general mapping and photogrammetric tasks. Sensor size, focal length and lens quality are the main camera factors that influence the accuracy of the survey. More advance remote sensing applications require the adoption of multispectral and hyperspectral sensors. The former encompasses both modified RGB cameras to near infrared and multispectral cameras with great prospects for precision agriculture, forestry and a broad range of vegetation studies [240]. Hyperspectral sensors collect information in multiple bands across the electromagnetic spectrum, and are of great interest to remotely observe the spectral response of many distinct biophysical parameters [22] and physiological process of organisms [124]. These families of sensors require radiometric calibration to account for variable lighting conditions and retrieve physical quantities that can be compared in time and with other sensors [241]. Thermal infrared cameras can remotely sense heat even in low visibility conditions and are ordinarily used for industrial inspection and surveillance, but also in soil science [242] and animal ecology [64]. Thermal sensitivity, expressed as the ability of the sensor to discriminate differences of temperatures even in low contrast scenes, is one of the most important technical aspects to increase the detection rate of wildlife [52]. LiDAR instruments are relatively expensive active sensors that can penetrate the canopy and derive accurate three-dimensional forest metrics and terrain models. However, structure-from-motion (SfM) [243] imaging techniques based on standard RGB cameras represent a low-cost alternative with limited, but reasonable results. In terms of platforms, long-endurance fixed-wing drones are preferred when surveying large areas and when landing is not a problem. Conversely, rotary-wing platforms are more versatile, and can operate in a diverse range of situations where precise flights prove more advantageous, such as in confined spaces and close-range inspection tasks, marine settings and terrestrial areas with steep terrain, or extensive vegetation cover.

**Table 4.** Suitable sensors for research and management tasks.

| Sensor | Applications |
| --- | --- |
| Visible RGB | Aerial photography, habitat mapping, photogrammetry, 3D Modeling, inspection, wildlife surveys (identification), landslides |
| Multispectral | Vegetation indices, productivity, water quality, geological surveys |
| Hyperspectral | Vegetation studies, biophysical variables, ecological processes, forest health, chlorophyll content, insect outbreaks. |
| Thermal | Inspection, wildlife surveys (detection), surveillance, wildfires, soil temperature, volcanology |
| LiDAR | 3D Modeling, topographical maps, forest inventory and metrics (structure, biomass, tree volume, canopy height, leaf area index) |

### 3.4. Knowledge Gaps and Recommendations for Future Research

The variety of information gathered from drones represents a great opportunity to complement ongoing Earth Observation programs aimed to monitor anthropogenic pressures threatening the ecological integrity of protected areas [244]. Drones can be rapidly deployed there, where early sign of disturbance have been previously detected using satellite images and environmental sensor networks [245]. Although many protected areas are too large to be mapped using drones, there are small, inaccessible and environmentally sensitive terrestrial and marine areas (ESAs) with important ecological values that could take advantage from drones. Once the use of drones has proven feasible in many different fields of application, it would be of interest that research focuses on methods to produce a set of ecological indicators in line with established monitoring frameworks [246]. For example, a wide range of biodiversity metrics, ecosystem processes and natural and anthropogenic stressors could be measured or derived, but further efforts are required to transfer advances on the field into accessible products for direct use at management levels. Table 5 suggest some potential challenges that can help to guide future research in the field.

**Table 5.** Challenges for the effective implementation of drones in protected areas.

| Management Categories | Challenges |
| --- | --- |
| Wildlife Research and Management | • Development of drones to minimize impact of wildlife.<br>• Optimization of automatic pattern recognition algorithms.<br>• Robust sampling design/limited statistical power.<br>• Integrating movement and visible/thermal data.<br>• Population structure and function, wildlife traits. |
| Ecosystem Monitoring | • Consistent ecological indicators.<br>• Multitemporal studies.<br>• Targeting Essential Biodiversity Variables (EBVs).<br>• Multiscale studies/linking drones with Earth Observation systems.<br>• Mapping of aquatic environments/bathymetry maps<br>• Machine learning methods (neural networks, etc.)<br>• Ecosystem services/area designation and performance.<br>• Habitat suitability/species reintroduction studies |
| Law Enforcement | • Research required to assess the performance of drones to reduce illegal activities.<br>• Test hybrid (VTOL) platforms.<br>• Marine Protected Areas: Drones/Vessel patrols<br>• Focus on poaching, but there are other important human intrusions in protected areas that could benefit from drones (illegal logging, mining, etc.)<br>• Threat maps. |
| Ecotourism | • Cost/benefit analysis<br>• Potential to introduce virtual flights.<br>• Fine-scale geofencing maps (Detailed map of sites where drone flights are allowed/conditioned/restricted) |
| Environmental Management and Disaster Response | • Move from prototypes to products and services.<br>• Implementation of Regional/Global Infrastructures for decision support.<br>• Satellite/Drone Remote Sensing integrative approach to model disturbance regimes. |

**Supplementary Materials:** The following are available online at http://www.mdpi.com/2504-446X/3/1/10/s1, **Figure S1.** PRISMA Flowchart; **Table S1**. List of studies.

**Author Contributions:** J.J.L. and M.M.P conceived and designed the study; J.J.L. collected and analyzed the studies; J.J.L. and M.M.P wrote the manuscript.

**Funding:** This research received no external funding.

**Acknowledgments:** The authors would like to thank the Editorial Office, and anonymous reviewers for their valuable comments and suggestions.

**Conflicts of Interest:** The authors declare no conflict of interest.

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
