# Peer review of "Drones for Conservation in Protected Areas: Present and Future"

_drones, doi:10.3390/drones3010010_

Round 1
Reviewer 1 Report
I appreciate the intent of this review and find the breadth of
relevant publications that you've found to be interesting. However, I
finished reading the ms wondering what exactly it contributes. It is not
an evaluation of the relative costs benefit of drones compared to other
flying platforms, such as fixed wing aircraft. Nor is it a a guide for
area managers to choose the best options for monitoring since there is
no information on the specific sensors applied for each task (see more
below). I'm not clear how the current ms helps "bridge the gap between
conservation science and management", or that it provides "a snapshot of
the current status" of drone use in conservation. Likely you have the
requisite information from your literature review and I would encourage
you to expand the scope and detail of your summaries.
Any RPAS is only a mobile platform carrying sensors. It is the type of sensor that will in large part determine the type of information that can be collected. It is important, therefore, to review the range of sensors available, as well as applied for different purposes, but also to introduce this idea early in the introduction.
A brief review of sensors, (and perhaps a summary table) would set up your summary of the literature, which is currently limited by not explicitly linking sensors to applications. For example, L122 "mapping of environmental correlates" needs more detail of which specific variables were targeted, and thus which sensors were used or are most appropriate. Part of your literature review could be to identify specific gaps or challenges - including , say, the lack of or costs of certain types of sensors that would allow expansion of critical research or monitoring.
Similarly,
you can't say (L140) "RPAS images" - the flying platform takes no
images; this depends entirely on the type and quality of the sensor
carried by the drone. In the case of vegetation mapping the comparison
should be between drones and helicopters or fixed wing aircraft as
platforms for photographic sensors, as each could carry the same
sensors, but each comes with its own costs and convenience. Consider too
that new high resolution satellite images give sub metre resolution
over large areas, so there is a trade off between image precision and
the post processing costs required to stitch together a series of photos
taken from a drone or other flying platform. Explicit cost comparisons
should be possible
The law enforcement section must consider
the different sensor options, for example optical image capture might
more constitute a breach of human rights than say, thermal imagery, yet
the latter is presumably useful for anti-poaching patrols?
You need a general summary table collating all your results, rather than relying on the text descriptions only. You make a nod towards managers seeking options for cost effectiove monitoring (L245), but unhelpfully suggest only that managers need to do "a trade off analyses among available platforms" - this could (should) have been a major focus of your review.
Author Response
Reviewer 1
I appreciate the intent of this review and find the breadth of relevant publications that you've found to be interesting. However, I finished reading the ms wondering what exactly it contributes. It is not an evaluation of the relative costs benefit of drones compared to other flying platforms, such as fixed wing aircraft. Nor is it a a guide for area managers to choose the best options for monitoring since there is no information on the specific sensors applied for each task (see more below). I'm not clear how the current ms helps "bridge the gap between conservation science and management", or that it provides "a snapshot of the current status" of drone use in conservation. Likely you have the requisite information from your literature review and I would encourage you to expand the scope and detail of your summaries.
Query 1
Any RPAS is only a mobile platform carrying sensors. It is the type of sensor that will in large part determine the type of information that can be collected. It is important, therefore, to review the range of sensors available, as well as applied for different purposes, but also to introduce this idea early in the introduction.
We have followed your suggestions and added a brief paragraph (L40) and two tables (table 1, 2) introducing RPAS platforms, sensors and other devices. For completeness of this section, we also cited some references that specifically review these important aspects. A new section was also included (L335) to briefly explain most suitable sensors according to the scope of application. A summary table (table 5) has been added for completeness.
Query 2
A brief review of sensors, (and perhaps a summary table) would set up your summary of the literature, which is currently limited by not explicitly linking sensors to applications. For example, L122 "mapping of environmental correlates" needs more detail of which specific variables were targeted, and thus which sensors were used or are most appropriate. Part of your literature review could be to identify specific gaps or challenges - including , say, the lack of or costs of certain types of sensors that would allow expansion of critical research or monitoring.
As previously noted, we followed your suggestion and added more information regarding sensors and applications. Last section (L363) and table 6 encompass specific gaps, challenges and recommended steps to be taken to expand research.
L122. We have modified the sentence accordingly to be more specific (L160). There are several examples along the manuscript that have been explicitly linked to sensors. Table 5
Query 3
Similarly, you can't say (L140) "RPAS images" - the flying platform takes no images; this depends entirely on the type and quality of the sensor carried by the drone. In the case of vegetation mapping the comparison should be between drones and helicopters or fixed wing aircraft as platforms for photographic sensors, as each could carry the same sensors, but each comes with its own costs and convenience. Consider too that new high resolution satellite images give sub metre resolution over large areas, so there is a trade off between image precision and the post processing costs required to stitch together a series of photos taken from a drone or other flying platform. Explicit cost comparisons should be possible
L140. We agree and modified the whole paragraph accordingly (L164). We also consider the trade-off between using satellite and RPAS for mapping. Particularly, we remarked the convenience of using RPAS for small surveys and specific tasks. Later we point out that the advent of long-endurance RPAS platforms can bring new opportunities to mapping greater extensions (L306).
Section 3.2.3 (L287) and 3.2.4 (L312) specifically tackles some economic and technological limitations of RPAS. We condense most of the limitations in these sections, but also made some comments inline were appropriate.
Table 4 (L310) provide some specific examples gathered from the literature were explicit comparison with established methods have been made. This table gather particular uses of RPAS in specific contexts.
Query 4
The law enforcement section must consider the different sensor options, for example optical image capture might more constitute a breach of human rights than say, thermal imagery, yet the latter is presumably useful for anti-poaching patrols?
We have followed your suggestions and modified the text accordingly (L218). This section has been revised to include some suggestions,
You need a general summary table collating all your results, rather than relying on the text descriptions only. You make a nod towards managers seeking options for cost effective monitoring (L245), but unhelpfully suggest only that managers need to do "a trade off analyses among available platforms" - this could (should) have been a major focus of your review.
Table 3 summarizes the studies conducted with RPAS according to the scope of application. We hope that the aggregated figures and tables gather most of the relevant points that we address throughout the manuscript.
L245. We have modified the sentence to better express the intended idea. Both sensors and platforms are introduced early in the manuscript. Later we linked sensors with application and provide a table summarizing these points. We hope that the information provided can help managers make convenient decisions in this regard.
Reviewer 2 Report
The ms is a review of peer-reviewed journal articles about drones used for research and management of protected areas (PAs). Its stated aim is to 'bridge the gap between conservation science and management' to 'help identify plausible scenarios for effective conservation'
Methodology is acceptable, though it is unusual to use Google Scholar and ResearchGate rather than databases that evaluate sources. The authors should explain how they ascertained peer review, and preferably should use Scimago or Web of Science to establish credibility of the sources they chose to keep. I suspect that doing so would reduce the number of articles somewhat.
As a review, the ms appears to be non-critical and does not offer a particularly new structure to available information. It will become outdated quickly as new research is published.
I would suggest moving information to tabular form where possible, so the reader can easily visualize the distribution of research.
Achieving the stated aim requires more than a literature review; especially an uncritical one. It requires obtaining feedback from PA managers to determine openness to the technology and reasons for / for not adopting the technology into routine tasks. Therefore, authors should modify the stated aim if they wish to resubmit.
Author Response
Reviewer 2
The ms is a review of peer-reviewed journal articles about drones used for research and management of protected areas (PAs). Its stated aim is to 'bridge the gap between conservation science and management' to 'help identify plausible scenarios for effective conservation'
Query 1
Methodology is acceptable, though it is unusual to use Google Scholar and ResearchGate rather than databases that evaluate sources. The authors should explain how they ascertained peer review, and preferably should use Scimago or Web of Science to establish credibility of the sources they chose to keep. I suspect that doing so would reduce the number of articles somewhat.
We have followed your suggestions as possible. We consulted Journal policies to assert peer-reviewed articles and remove conference proceedings that were not published in peer-review journals. Thus, the list of articles selected was reduced to 256. We agree that Google Scholar and Research Gate might not be the best tools to do a bibliographical search, but the procedure can be reasonably reproduced if care and large doses of patience are taken.
Query 2
As a review, the ms appears to be non-critical and does not offer a particularly new structure to available information. It will become outdated quickly as new research is published.
We acknowledge that the RPAS field is rapidly evolving and the industry is moving fast. These factors make challenging to write an up-to-date literature review. Notwithstanding, we consider that the manuscript can prove relevant for many readers, because it condenses different aspects of RPAS for conservation and management into a single document.
Query 3
I would suggest moving information to tabular form where possible, so the reader can easily visualize the distribution of research.
We have followed your suggestions. There are several new tables (table 1-6) and figures (1A 1B 1C) that can help to visualize the distribution of research.
Query 4
Achieving the stated aim requires more than a literature review; especially an uncritical one. It requires obtaining feedback from PA managers to determine openness to the technology and reasons for / for not adopting the technology into routine tasks. Therefore, authors should modify the stated aim if they wish to resubmit.
We generally agree and modified the stated aim according to the scope and limitations of the manuscript (L78).
Reviewer 3 Report
I generally like the review and I think it would make a nice and important contribution to the scientific community. However, the manuscript could gain from a language revision and editing to improve its readability. In addition, the structure of the paper could be improved by making a clear separation between results and discussion. The results section needs more added value information such as ranking, graphs, tables… as well as evidence of the knowledge gained from the review (potential gaps…)
Author Response
Reviewer 3
I generally like the review and I think it would make a nice and important contribution to the scientific community.
Query 1
However, the manuscript could gain from a language revision and editing to improve its readability.
We agree and followed your suggestions accordingly. The manuscript has undergone significant changes. We first carried out an in-depth language revision. Later the manuscript was thoroughly checked by a native English speaker. Simultaneously, many sentences along the text were reformulated and adapted. We consider that the manuscript has benefit from strong editing and the readability and clarity of the message has improved.
Query 2
In addition, the structure of the paper could be improved by making a clear separation between results and discussion.
We appreciate the suggestion and indeed considered this structure in early drafts, but the literature review had become too extensive and repetitive. Since there are similar peer-reviewed articles that does not follow the classical structure, we opted for relax these rules. Nevertheless, the MDPI journal polices stated that for review manuscripts only front matter (Title, Author list, Affiliations, Abstract, Keywords), literature review sections and the back matter (Supplementary Materials, Acknowledgments, Author Contributions, Conflicts of Interest, References) sections are mandatory.
Query 3
The results section needs more added value information such as ranking, graphs, tables… as well as evidence of the knowledge gained from the review (potential gaps…)
We have followed your suggestions and added several figures (figure 1A 1B 1C) and tables (table 1-6) accordingly. Specifically, a new section has been written (L363) to address knowledge gaps and recommendations for future research. Table 6 summarizes some specific challenges that we believe should be considered in future research.
Round 2
Reviewer 1 Report
Thank you for your careful revision
Author Response
Your suggestions are highly appreciated.
Reviewer 2 Report
Manuscript is much improved, particularly the addition of artwork.
Language is still awkward in places, though overall the ms is much easier to read now. Specific issues are below, but there are many more changes that would improve readability
9 threaten
45 change 'with respect not long ago"
58 I think you mean 'passive'
144 significative
182 notoriously - poor word choice
227 effectiveness?
275 pertubation?
276 exempt from discussion,
279 most studies
282 weighed
290 hardly - I think this word has the opposite meaning to your intention
303 they present
326 cope?
376 could potentially
Author Response
Thank you very much for your help and the care you have shown for the details. We have corrected most spelling and grammar problems according to your indications. We hope that the editors can help us solve other related issues that we could not detect.
9 threaten.
We fixed the spelling.
45 change 'with respect not long ago"
We changed the sentence and use “with respect previous photogrammetric techniques” instead.
58 I think you mean 'passive'
We fixed the spelling.
144 significative
We changed the word and use “significant”.
182 notoriously - poor word choice
We used “noticeably” instead.
227 effectiveness?
We used the word “performance” to avoid repetition.
275 pertubation?
We used “alteration” instead.
276 exempt from discussion,
Removed to avoid repetition.
279 most studies
Corrected
282 weighed
Corrected
290 hardly - I think this word has the opposite meaning to your intention
We used the word “difficult”.
303 they present
The subject was included.
326 cope?
We used the word “prevail” instead.
376 could potentially
We used just “could”